# Hybrid Approach for Optimisation and Analysis of Palm Oil Mill

**Steve Z. Y. Foong [1], Viknesh Andiappan [2], Raymond R. Tan [3], Dominic C. Y. Foo [1] and Denny K. S. Ng [1,2,\*]**

[1] Department of Chemical and Environmental Engineering, The University of Nottingham Malaysia Campus, Broga Road, 43500 Semenyih, Malaysia; kebx6fzy@exmail.nottingham.edu.my (S.Z.Y.F.); Dominic.Foo@nottingham.edu.my (D.C.Y.F.)

[2] School of Engineering and Physical Sciences, Heriot-Watt University Malaysia, 62200 Putrajaya, Wilayah Persekutuan Putrajaya, Malaysia; v.murugappan@hw.ac.uk

[3] Centre for Engineering and Sustainable Development Research, De La Salle University, 2401 Taft Avenue, 0922 Manila, Philippines; raymond.tan@dlsu.edu.ph

\* Correspondence: Denny.Ng@hw.ac.uk; Tel.: +60-3-8894-3784

**Abstract:** A palm oil mill produces crude palm oil, crude palm kernel oil and other biomass from fresh fruit bunches. Although the milling process is well established in the industry, insufficient research and development reported in optimising and analysing the operations of a palm oil mill. The performance of a palm oil mill (e.g., costs, utilisation and flexibility) is affected by factors such as operating time, capacity and fruit availability. This paper presents a hybrid combined mathematical programming and graphical approach to solve and analyse a palm oil mill case study in Malaysia. The hybrid approach consists of two main steps: (1) optimising a palm oil milling process to achieve maximum economic performance via input-output optimisation model (IOM); and (2) performing a *feasible operating range analysis* (FORA) to study the utilisation and flexibility of the developed design. Based on the optimised results, the total equipment units needed is reduced from 39 to 26 unit, bringing down the total capital investment by US$6.86 million (from 18.42 to 11.56 million US$) with 23% increment in economic performance (US$0.82 million/y) achieved. An analysis is presented to show the changes in utilisation and flexibility of the mill against capital investment. During the peak crop season, the utilisation index increases from 0.6 to 0.95 while the flexibility index decreases from 0.4 to 0.05.

**Keywords:** mathematical programming; graphical approach; feasible operating range analysis; utilisation index; flexibility index

## 1. Introduction

Oil palm (*Elaeis guineensis*) is cultivated for the production of fresh fruit bunches (FFBs) due to its stability, high yield and low cost [1,2]. FFBs are then can be converted into a variety of products including foods, cosmetics, detergents and biofuels. To date, approximately 85% of global crude palm oil (CPO) is produced in Indonesia and Malaysia [3]. CPO is extracted from FFBs in processing facilities known as palm oil mills (POM). A typical milling process consists of several operational units as shown in Figure 1. FFBs undergo sterilisation, threshing, digestion and pressing to produce pressed liquid and cake. The pressed liquid is clarified and purified to produce CPO, while the pressed cake undergoes nut separation, nut cracking, kernel separation and drying to produce palm kernel (PK). Most POMs in Malaysia will send the PK to a kernel crushing plant for crude palm kernel oil (CPKO) production [4] before refinery processes where CPO and CPKO are refined into higher quality edible

oils and fats [5]. Throughout the milling process, biomass such as palm kernel shell (PKS), pressed empty fruit bunch (PEFB) and palm pressed fibre (PPF) are generated as by-products. Meanwhile, large amounts of strong wastewater, which is known as palm oil mill effluent (POME) are produced during sterilisation and clarification operations.

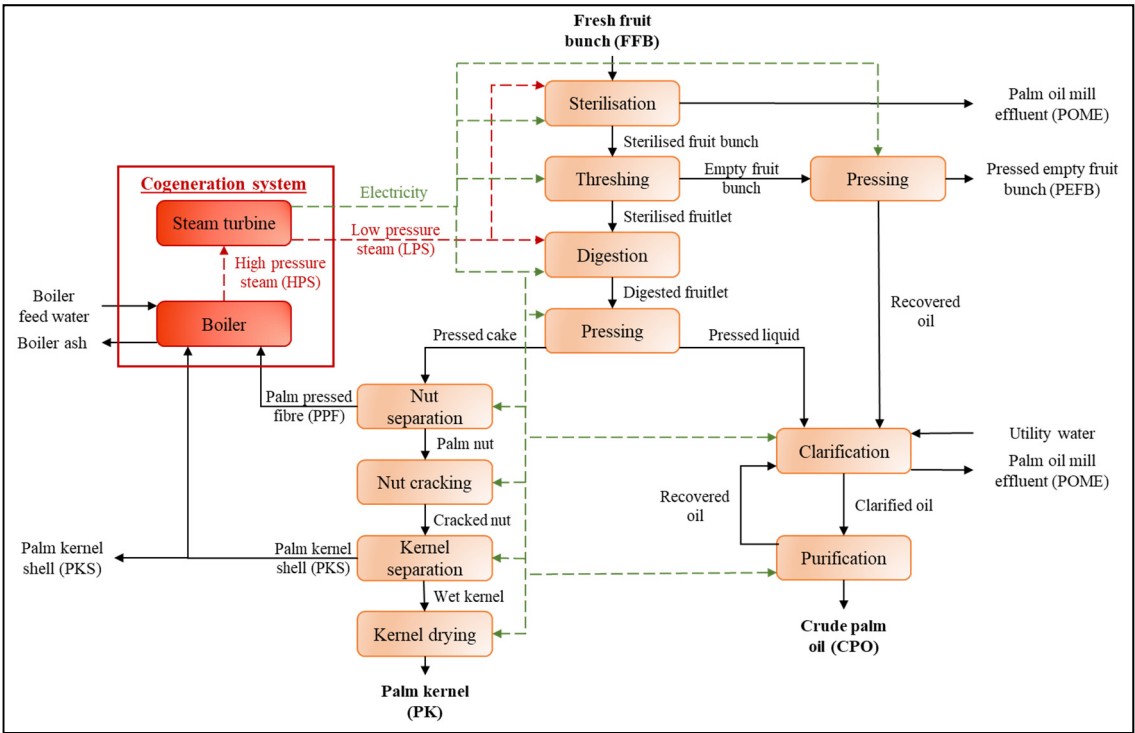

**Figure 1.** Unit operations in a typical palm oil mills (POM) [6].

POMs are usually located near to the plantations, which usually are in remote areas to minimise logistics costs. In Malaysia, 63% of the active POMs are positioned far away (>10 km) from electrical grid connection point [7], leaving them at a disadvantage as they would require steam and electricity for CPO extraction. Abdullah and Sulaiman [8] estimated that 0.075–0.1MWh electricity and 2.5 t of low-pressure steam (LPS) are required per ton of CPO produced. In current practice, over three-quarters of over 400 POMs in Malaysia met the process steam and electricity demands by burning PPF and a portion of PKS generated from the milling process [9,10] via co-generation [11]. Excess PKS can then be sold as an alternative solid fuel around the world [12,13], while PEFB is returned to plantations as mulching materials [14] or composted to produce biofertilizer [15]. The biomass can also be used for a range of other applications (e.g., pellet, dried long fibre, etc.). Meanwhile, pond-based wastewater treatment systems are commonly used to treat POME before discharge [16].

Yu-Lee [17] stated that the processing capacity of a plant or system depends on the labour, equipment, technology and materials available. In this sense, POMs would have their unique design features and the operations of each mill may differ between one another. For instance, the capacity of a typical POM could range between 20 to 90 t/h of FFB, with operations up to 19 h every day [18]. Besides, ripe FFBs collected from plantations must be transported and processed immediately in POMs to prevent degradation of CPO quality due to increased free fatty acid content [19]. The amount of FFBs supplied to a POM could vary depending on location and time, due to seasonal crop changes and possible unforeseen circumstances in the plantations [6,20]. To overcome these issues, most plants or systems including POM are often built with an excess capacity to ensure higher flexibility [21] and lower processing costs (i.e., labour, service and maintenance costs) [22]. However, this affects the utilisation and economic performance of POM, especially during the lean crop season.

According to the literature, there are several methods developed to optimise and analyse the performance of systems; one of the commonly used methods is input-output (IO) model. IO model was first developed by Leontief [23] to deal with the interdependencies between system components (e.g., materials, processes, costs) using systems of linear equations. IO models are used to study the behaviour of a system when the input or output of one system component changes quantitatively [24]. Some notable works on IO model have been presented to analyse economic networks [23], industrial networks [25], chemical industry supply chains [26], food manufacturing plants [27] and life cycle assessment [28,29]. IO optimisation models (IOM) have also been developed based on the general IO methodology. IOM has been successfully applied for industrial complexes [30,31], biorefineries [32], sustainable industrial systems [33], human resources [34] and palm oil plantations [35] to make the best use of situation, goods or production capacity.

Apart from IOM, graphical approaches have been developed to analyse system performance. Graphical approaches provide visual assistance in analysing scientific data and communicating quantitative information [36]. Some of the well-known graphical approaches are the insight-based *pinch analysis technique* [37] and *process graph*, also known as P-graph [38]. Detailed information and applications of such approaches have been reviewed and discussed by Linnhoff [39], Foo [40], and Teng et al. [41]. Recently, Andiappan et al. [42] proposed the *feasible operating range analysis* (FORA) to examine the real-time feasible operating range of an energy system graphically. Such approach allows the range output (i.e., maximum and minimum of each output) of a system to be determined, considering material input and capacity constraints of individual unit operations. Besides, it also provides insight into potential design modifications based on variations in output demand and process bottleneck [43].

The studies presented thus far provide evidence for the applications of mathematical programming and graphical approaches (i.e., IOM and FORA) to optimise and analyse problems in various fields. However, limited works were reported for a hybrid approach to deal with such issues. None of the contributions discussed has focused on palm oil milling processes apart from Foong et al. [6], in which a mathematical programming approach alone is presented. Based on the previous work [6], operational variables such as operating hours and labour costs are yet to be considered. Besides, analysis on a real-time feasible operating range and the bottleneck of the developed design is not performed in the previous work. In addition, the operational performance of the milling process can be quantified in terms of utilisation and flexibility indices, introduced by Grossmann et al. [44] to measure the usage and expected deviation from a nominal design state that a process can handle. These research gaps are dealt with in this study, developing a hybrid approach consisting of IOM for palm oil mill optimisation, followed by FORA to analyse the feasible operating range of the developed system. In particular, this work provides an extended account of FORA, whereby production rates, flexibility and utilisation indices and capital expenditure are considered simultaneously to provide a visualisation tool for process improvement.

In the following section, the problem statement for this work is presented, followed by a detailed formulation for IOM in Section 3. Next, an existing POM flowsheet is optimised using the input-output approach described in Section 4. Following this, the economic performance, utilisation and flexibility of the POM are then compared to highlight the improvements achieved. Lastly, the conclusions and prospective future works are described in the final section.

## 2. Problem Statement

The problem addressed by the proposed approach is divided into two parts, stated as follows. The palm oil milling processes consist of a set of technology $te \, \epsilon$ TE with interchangeable material $m \, \epsilon$ M. Firstly, an IOM is developed where **A** is the input and output matrix composed of the fixed interaction ratios, $a_{m,te}$ between material $m$ and technology $te$. Each crop season $s$ has a fixed fraction of occurrence, $\alpha_s$, to indicate the proportion of each year that it takes up. Different levels of supply of material $m$ are available in each crop season $s$. The number of equipment units operated, $U_{te}$

determined from the nominal capacity, $\mathbf{CAP}_{te}$ available in the market. Each material $m$ and technology *te* associated with a given material cost, $\mathbf{C}_m$, operating cost, $\mathbf{OC}_{te}$, capital cost, $\mathbf{CC}_{te}$ and electricity consumption, $\mathbf{E}_{te}$, respectively. In the event where annual operating time, *AOT* exceeds the annual shift time, AST, additional overtime cost, *OTC* and operating costs, *OPEX* required. The objective is to maximise the economic performance, *EP* of the POM as shown in Equation (1).

$$\text{Maximise } EP \tag{1}$$

Based on the optimised POM design, the $\boldsymbol{U}_{te}$ determined is set as the maximum units operated, $\mathbf{U}_{te}^{\max}$ to identify the technology bottleneck, $B_{te}$ from the maximum capacity, $\mathbf{CAP}_{te}^{\max}$ of each technology *te*. Next, FORA is then performed to evaluate the developed system using utilisation and flexibility indices, *UI* and *FI*, respectively. The following section further explains the approach developed for this work.

## 3. Hybrid Approach Formulation

As mentioned previously, a hybrid approach is developed in this work to optimise the palm oil milling process via IOM, followed by FORA to analyse the developed system. The italic notations represent the variables determined by the model and non-italic notations represent constant parameters defined in the proposed approach. Meanwhile, matrix and vector symbols are represented by bold notations.

### 3.1. Input-Output Optimisation Model (IOM)

In this model, each crop season in which material flows would vary is represented by index $s$. It is assumed that a linear correlation for material flows in the milling process is given in Equation (2)

$$\mathbf{A}(\boldsymbol{x}_{te})_s = (\boldsymbol{y}_m)_s \ \forall m, \qquad \forall s \tag{2}$$

where $\mathbf{A}$ is the matrix consists of fixed interaction ratios, $a_{m,te}$ for material input and output ratios, to and from technology *te*. Each column in matrix $\mathbf{A}$ corresponds to different technology *te*, while its rows correspond to material $m$ flows. $a_{m,te}$ are expressed in negative values for material inputs, positive values for material outputs or zero if there are no interactions between material $m$ and technology *te*. $\boldsymbol{x}_{te}$ is the processing capacity vector of technology *te*, in which positive values obtained for technologies operated and zero when it is not. Meanwhile, $\boldsymbol{y}_m$ is the flow rate vector of material $m$ (i.e., input or output). Final and by-products are indicated with positive values while process feedstocks are indicated with negative values and intermediates denoted with zeros. Note that both $\boldsymbol{x}_{te}$ and $\boldsymbol{y}_m$ are expressed in material flow rate (t/h) or power generation (kW).

In the process, electricity is also being consumed to operate technology *te* for material conversions. However, electricity demand, $E^{\text{Demand}}$ of a POM relies on the number of units operated for technology, $\boldsymbol{U}_{te}$ rather than linear correlation as shown in Equation (3).

$$\left(E^{\text{Demand}}\right)_s = \sum_{te=1}^{TE} (\boldsymbol{U}_{te})_s \mathbf{E}_{te} \qquad \forall s \tag{3}$$

$\mathbf{E}_{te}$ is a diagonal matrix for electricity consumption specified per unit technology *te* operated. Vector for the number of units of technology operated, $\boldsymbol{U}_{te}$ is determined based on the inverse of a nominal capacity diagonal matrix, $\mathbf{CAP}_{te}$ available in the market ($\mathbf{CAP}_{te}^{-1}$) obtained from Equation (4).

$$(\boldsymbol{U}_{te})_s \geq (\boldsymbol{x}_{te})_s \mathbf{CAP}_{te}^{-1} \ \forall s, \qquad \forall te \tag{4}$$

$\boldsymbol{U}_{te}$ consists of positive integers and the inequality in Equation (4) ensures that the products of $\boldsymbol{U}_{te}$ and $\mathbf{CAP}_{te}$ to be greater or equal to $\boldsymbol{x}_{te}$ for the process to operate.

In the presence of power supply from grid connection, the system produces and utilises electricity generated onsite. To ensure that the process is self-sufficient without interruption, an additional constraint, Equation (5) is included whereby the output of electricity produced, $y_{\text{electricity}}$ in the process is greater or equal to the electricity demand, $E^{\text{Demand}}$ in each crop season $s$.

$$\left(y_{\text{electricity}}\right)_s \geq \left(E^{\text{Demand}}\right)_s \qquad \forall s \tag{5}$$

Note that the focus of this work is to model the interdependency of each equipment with one another in a single system or plant. For conservative measurement, the power consumption and process efficiency for maximum loading is assumed for each operating equipment to prevent underestimation of power demand needed, regardless of the process throughput for each equipment. Every technology unit $te$ is sized based on these conservative values to ensure the reliability of system developed. As such, every time an equipment is selected, a conservative energy consumption value (or maximum) is activated.

Meanwhile, the economic performance, $EP$ of the process is evaluated based on Equation (6)

$$EP = GP - CRF \times CAPEX \tag{6}$$

where $GP$, $CRF$ and $CAPEX$ represent the gross profit, capital recovery factor and capital costs required, respectively. To ensure that the developed system can sustain itself economically, $EP$ must be greater or equal to zero. Next, Equation (7) is used to calculate $GP$

$$GP = \sum_s \alpha_s \left[ \left( AOT \sum_{m=1}^{M} \boldsymbol{y}_m \mathbf{C}_m - OPEX - OTC \right)_s - LC \right] \tag{7}$$

whereby $AOT$, $\alpha_s$, $\mathbf{C}_m$, $OPEX$, $OTC$ and $LC$ are the annual operational time, fraction of occurrence, material, total operating, overtime and labour costs, respectively. Equation (7) is subject to

$$\sum_s \alpha_s = 1 \tag{8}$$

in which the inclusion of $\alpha_s$ assessed the performance of the system developed in all crop season $s$. Each fraction of occurrence represents the time fraction where a season occurs. The summation of these fractions must equal to one as shown in Equation (8) as the time fraction is obtained by dividing the duration of a crop season $s$ with the total duration considered. $AOT$ is determined by Equation (9)

$$\left(m^{\text{max}}\right)_s \geq \left(AOT \times y_m\right)_s \qquad \forall s \tag{9}$$

where $m^{\text{max}}$ is the maximum material demand (positive value) or available (negative value) per annum, depending on the constraint set for each season $s$. Equation (9) is subject to

$$\left(AOT\right)_s \leq AOT^{\text{max}} \qquad \forall s \tag{10}$$

where $AOT^{\text{max}}$ is the maximum annual operating time of the process.

$CRF$ is used to annualise $CAPEX$ over a specified operation lifespan $t_{te}^{\text{max}}$ and discount rate, r, determined via Equation (11).

$$CRF = \frac{r\left(1 + r\right)^{t_{te}^{\text{max}}}}{\left(1 + r\right)^{t_{te}^{\text{max}}} - 1} \tag{11}$$

$CAPEX$ is calculated based on the units of technology installed during the high crop season, $(\boldsymbol{U}_{\text{te}})_{\text{H}}$ while $OPEX$ depends on the units of technology operated, $\boldsymbol{U}_{te}$ in the process as shown in Equations (12)–(13).

$$CAPEX = \sum_{te=1}^{TE} (\boldsymbol{U}_{te})_{\text{H}} \mathbf{CC}_{te} \tag{12}$$

$$(OPEX)_s = \sum_{te=1}^{TE} (\boldsymbol{U}_{te})_s \boldsymbol{OC}_{te} \ \forall s \tag{13}$$

$\boldsymbol{CC}_{te}$ and $\boldsymbol{OC}_{te}$ represent the capital and operating costs per unit of technology *te*, expressed in diagonal matrixes. Meanwhile, Equations (14) and (15) determine *OTC* and *LC* required.

$$(OTC)_s = C_{OT} n_{wk} [(AOT)_s - AST] \qquad \forall s \tag{14}$$

$$LC = C_{lab} n_{wk} n_{ws} \tag{15}$$

where $C_{OT}$ and $C_{lab}$ are the specific overtime cost and labour cost; $n_{wk}$ and $n_{ws}$ represent the number of workers and working shifts per day; AST is the annual shift time of the process.

*3.2. Feasible Operating Range Analysis (FORA)*

It is worth mentioning that the optimal design obtained using IOM is only optimised for a given set of conditions. When changes arise in the near future, it is important to have sufficient flexibility to cater for such changes. As such, FORA provide a clear visualisation to avoid the system developed from over- or under-designed. In fact, it provides flexibility for the decision maker to decide on the required design flexibility based on how much *CAPEX* to be invested. Based on the IOM developed previously, FORA is performed to analyse the feasible operating range of the POM designed. The analysis begins by setting the maximum units of technology installed, $\boldsymbol{U}_{te}^{max}$ as the $\boldsymbol{U}_{te}$ of the design with the smallest capacity (i.e., during low crop season) as given in Equation (16).

$$\boldsymbol{U}_{te}^{max} = (\boldsymbol{U}_{te})_L \tag{16}$$

The product of $\boldsymbol{U}_{te}^{max}$ and $\boldsymbol{CAP}_{te}$ gives the maximum capacity, $\boldsymbol{CAP}_{te}^{max}$ as shown in Equation (17) and the inverse matrix, $(\boldsymbol{CAP}_{te}^{max})^{-1}$ is used in Equation (18) to identify the technology bottleneck, $\boldsymbol{B}_{te}$ of the system. $\boldsymbol{B}_{te}$ ranges from zero to one where zero indicating that technology *te* is not utilised, while one shows the bottleneck of the entire system in which the capacity of that particular technology *te* is utilised to its maximum potential.

$$\boldsymbol{CAP}_{te}^{max} = \boldsymbol{U}_{te}^{max} \boldsymbol{CAP}_{te} \qquad \forall te \tag{17}$$

$$(\boldsymbol{x}_{te})_s (\boldsymbol{CAP}_{te}^{max})^{-1} = (\boldsymbol{B}_{te})_s \qquad \forall s, \forall te \tag{18}$$

In this work, the milling process is optimised with the objective function given in Equation (1) by deactivating the material input constraint, Equation (8) to determine the maximum product output of the system, $y_m^{max}$. At this point, the technology bottleneck of the system is indicated by $\boldsymbol{B}_{te}$ equal to one ($\boldsymbol{B}_{te} = 1$), representing that a particular technology has been fully utilised, capping the $\boldsymbol{y}_m$ of the entire system. It is assumed that process intensification of the technology bottleneck is not possible and additional equipment unit will be needed to increase $y_m^{max}$, where $\boldsymbol{B}_{te}$ serves as an indicator to pinpoint the additional technology equipment for purchase/upgrade.

Following that, the objective function is modified into Equation (19) to determine the minimum output of the system, $y_m^{min}$ while ensuring the system is economically stable to sustain its operation (i.e., *EP* equal to zero). In the event where minimum *EP* is required at a targeted value, additional constraints may be added to the formulation. The changes in $y_m^{max}$ and $y_m^{min}$ are measured for each incremental step in summation of $\boldsymbol{U}_{te}^{max}$ ($\sum_{te=1}^{TE} \boldsymbol{U}_{te}^{max}$) by one equipment unit at a time to determine the feasible operating range of each design.

$$\text{Minimise } EP \tag{19}$$

The utilisation index, *UI* and flexibility index, *FI* for each incremental unit of $\mathbf{U}_{te}^{\max}$ is determined via Equations (20) and (21) to measure the operational performance of the system

$$(UI)_s = \frac{(y_m)_s}{y_m^{\max}} \qquad \forall s \qquad (20)$$

$$(FI)_s = \frac{y_m^{\max} - (y_m)_s}{y_m^{\max}} \qquad \forall s \qquad (21)$$

where *UI* and *FI* range between zero to one. In the event where *UI* equals to zero, the process is not utilised while *UI* equals to one indicates that the process is operating at 100% of the processing capacity installed. Meanwhile, zero in *FI* represents that the process has no flexibility in its operation and vice versa. To better illustrate the proposed FORA, a generic process where $y_m^{\max}$, $y_m^{\min}$, *UI* and *FI* are plotted against *CAPEX* as shown in Figure 2a. Several key features to be highlighted from the analysis are as follows:

1.  Cross and plus markers in Figure 2a represent $y_m^{\max}$ and $y_m^{\min}$ of different system design with different $\mathbf{U}_{te}^{\max}$ (x-axis on the left) and a corresponding *CAPEX* required (y-axis). The area shaded in grey between $y_m^{\max}$ and $y_m^{\min}$ represented the feasible operating range of the developed system where $y_m$ (yellow line) must fall in between this region. This is to ensure that the system output is always less than or equal to the maximum production capacity, while greater or equal to the minimum output to sustain its operation.
2.  $y_m^{\max}$ and $y_m^{\min}$ changes with the system design, during the addition or removal of the equipment unit. Hence, step changes are observed in $y_m^{\max}$ and $y_m^{\min}$ (black lines) when *CAPEX* increases.
3.  The technology bottleneck and additional equipment unit to be added for each step are identified by $\boldsymbol{B}_{te}$ when $\boldsymbol{B}_{te} = 1$. Increments in $y_m^{\max}$ and $y_m^{\min}$ are not proportional to the increment in *CAPEX* as the capital and capacity of each technology varies according to the market. Occasionally, more than one technology bottlenecks might occur. In that respect, multiple types of equipment and greater *CAPEX* are needed to increase the capacity of the system.
4.  The increment in $y_m^{\max}$ (black line) reduces the *UI* (green line) while increases the *FI* (blue line) of the system in the same behaviour due to the changes in production capacity as shown in the x-axis on the right.

In the event where $y_m$ is increased to $y_m^{\max}$, the first design becomes infeasible (area shaded in red) as $y_m^{\text{new}}$ falls out of the area shaded in grey as shown in Figure 2b. At the same time, a budget constraint is applied where only an increment up to a maximum capital cost, CAPEX$^{\max}$ can be invested. Based on the diagram, design 2 is required to cope up with such changes and *CAPEX$_2$* falls within the constraint (*CAPEX$_2$* < CAPEX$^{\max}$). As such, the expansion from first to second design is feasible and the system will have more flexibility in product output up to $y_{m2}^{\max}$ level. On the other hand, when $y_m^{\text{new}}$ is further increased as shown in Figure 2c, the *CAPEX* required falls beyond the constraint (*CAPEX$_3$* > CAPEX$^{\max}$). This shows that none of the design is suitable for such increment in $y_m$. Hence, the decision maker may only expand up to $y_{m2}^{\max}$ with *CAPEX$_2$*, sacrificing the additional output between $y_m^{\text{new}}$ and $y_{m2}^{\max}$.

Besides targeting for changes in $y_m$, the proposed approach allows for decision making based on *FI* or *UI*. For instance, the same investor is interested in changing the system slightly, allowing for more flexibility, $FI^{new}$ to tolerate greater fluctuation in $y_m$ which may occur in the future. Similarly, budget constraint is capped at $CAPEX^{max}$. As presented in Figure 2d, $FI^{new}$ is greater than $FI_1$ but lesser $FI_2$. Therefore, flexibility up to $FI_2$ can be achieved based on the constraint set. This analysis serves as a powerful tool to plan for increment in product output expected in the future, under different constraints.

The following section presents a typical POM case study in Malaysia to illustrate the proposed approach. An IOM is developed to optimise the milling process, followed by FORA to study the feasible operating range within its design capacity. The developed mixed-integer linear programming model was solved via LINGO (v16, LINDO Systems, Inc., Chicago, USA) to achieve a global solution [45], with an Intel® Core™ i5 (2 × 3.20 GHz), 8 GB DDR3 RAM desktop unit. Alternatively, other optimisation software such as MATLAB and Statistics Toolbox (Release 2012b, The MathWorks, Inc., Natick, MA, USA) and General Algebraic Modeling System (GAMS) (Release 24.2.1, GAMS Development Corporation, Washington, DC, USA) could be used to achieve the same solution, depending on user preference.

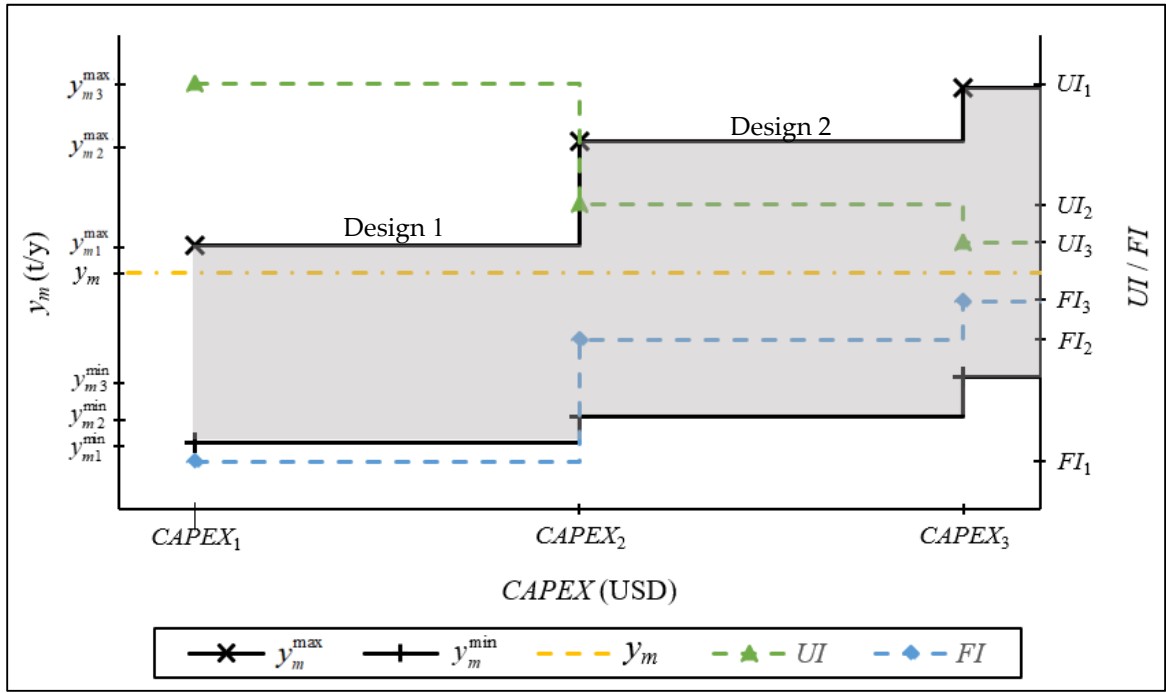

(**a**)

**Figure 2.** *Cont.*

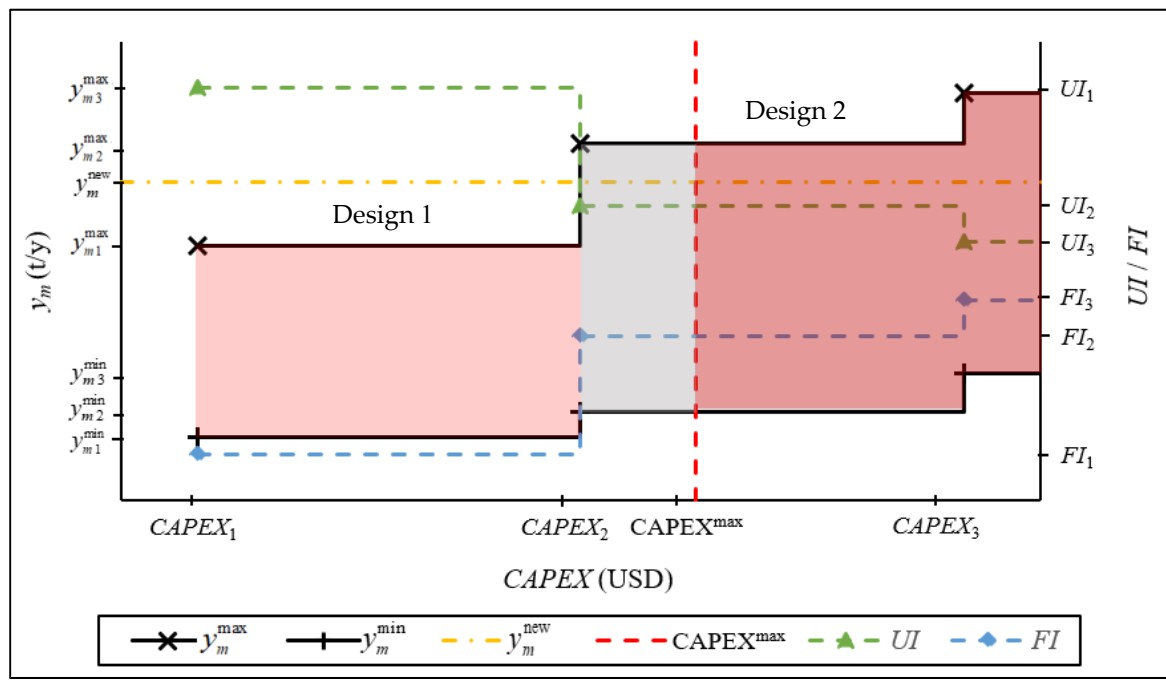

(**b**)

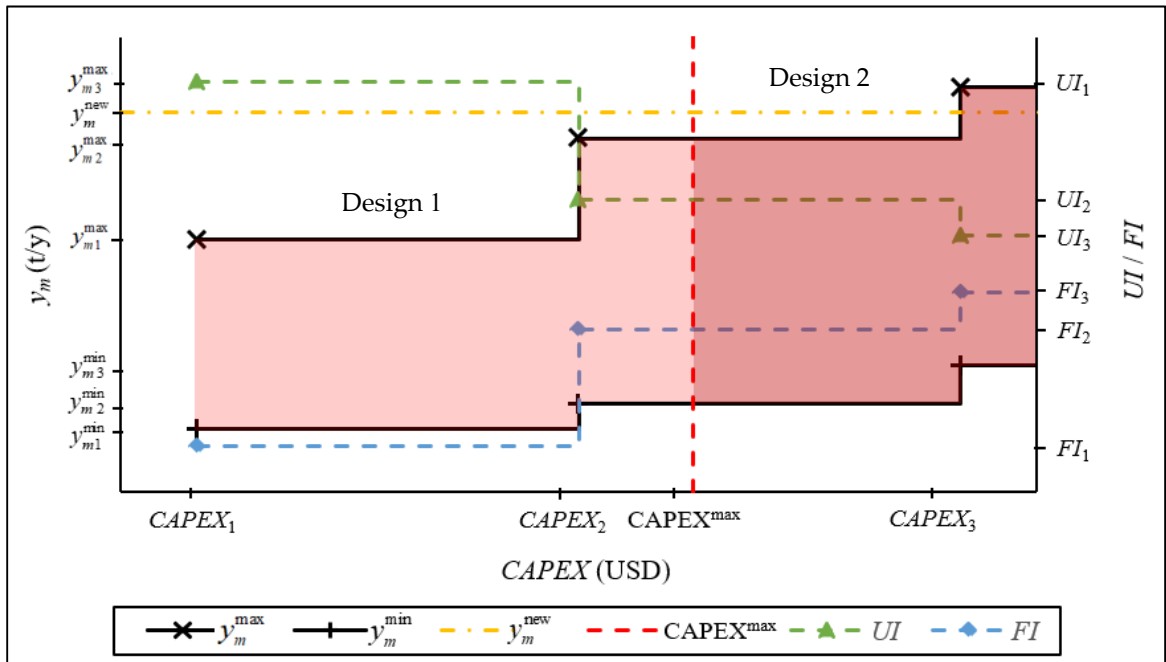

(**c**)

**Figure 2.** *Cont.*

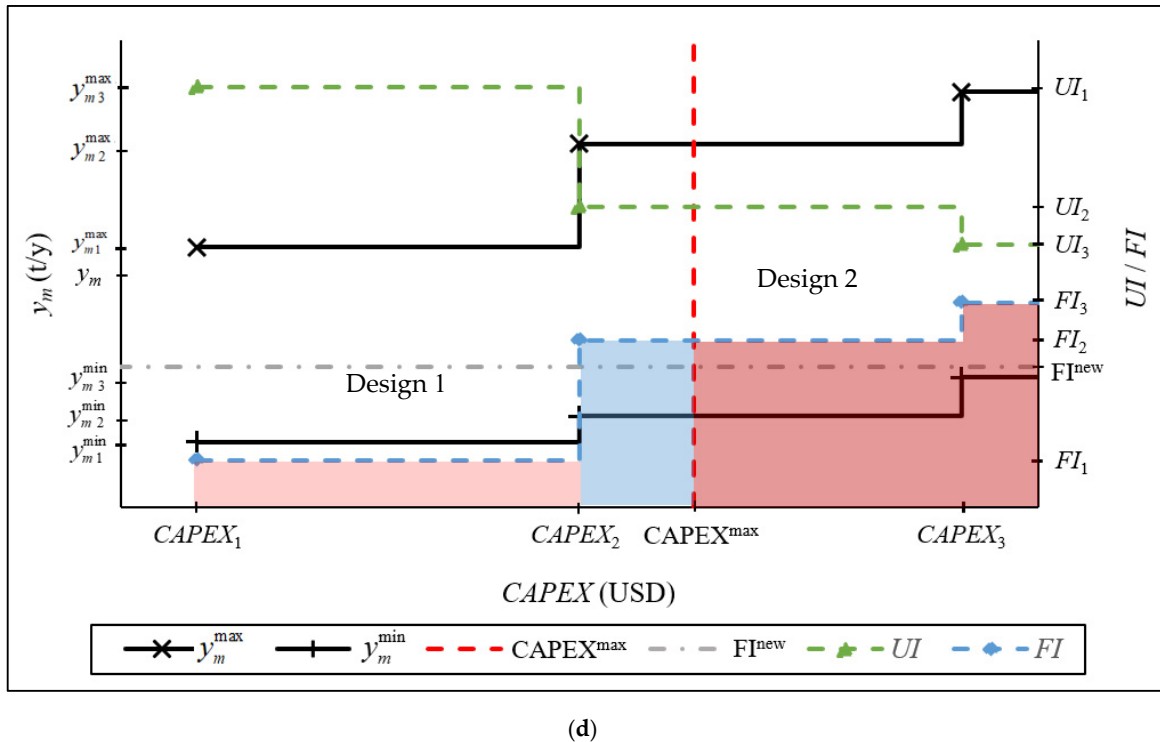

(**d**)

**Figure 2.** (**a**) FORA for a generic process. (**b**) FORA with $y_m^{new}$ within the CAPEX$^{max}$ constraints. (**c**) FORA with $y_m^{new}$ exceeding the CAPEX$^{max}$ constraints. (**d**) FORA with FI$^{new}$ and CAPEX$^{max}$ constraints.

## 4. Case Study

In this case study, the hybrid approach presented is demonstrated using a POM design adopted from Foong et al. [6] as the baseline design. It is assumed that the mill operator is interested to further optimise the milling process to improve economic performance, *EP* by taking operational factors such as operating hours, labour costs and FFB availability into account. Besides, an analysis to study the feasible operating range, utilisation and flexibility of the POM design is performed, providing a better insight for any changes in system design to cater for any variation in production output in the future. FFBs obtained from plantations are divided into three crop seasons, that is, low, medium and high seasons, each with a given fraction of occurrence, $\alpha_s$ and availability, $y_{FFB}$ as shown in Table 1.

**Table 1.** The fraction of occurrence and FFB availability for different crop seasons.

| Crop Season | Fraction of Occurrence, $\alpha_s$ | FFB Availability, $y_{FFB}$ (t/y) |
|---|---|---|
| Low | $\alpha_L = 0.417$ | $-195,800$ |
| Medium | $\alpha_M = 0.333$ | $-261,000$ |
| High | $\alpha_H = 0.250$ | $-369,800$ |
| Average | | $-261,000$ |

Reprinted (adapted) with permission from [6], copyright (2018) American Chemical Society.

A typical POM operates in batches for 12 h daily, usually divided into two workings shifts (i.e., annual shift time, AST = 4350 h/y). It is assumed that the POM is located in a remote area where power grid connection is not available and electricity required to operate the milling process is produced by cogeneration of biomass resources such as PPF and PKS. Fifteen operators with a labour cost, $C_{lab}$ of US\$4500/y is required for each shift to operate the milling process. It is further assumed that the POM will have an operation lifespan, $t_{te}^{max}$ of 15 years with a discount rate, r of 5% per annum. The baseline POM design is shown in Figure 3 with the material and energy flows reported in a range

for low and high crop seasons while the values stated in bracket represents the equipment units of each technology needed. Economic parameters such as *CAPEX, OPEX, LC, GP* and *EP* are summarised in Table 2 while additional information on material flows, technology units, process matrix table and other specifications of the system (i.e., $\mathbf{CAP}_{te}$, $\mathbf{E}_{te}$, $\mathbf{CC}_{te}$, $\mathbf{OC}_{te}$ and $\mathbf{C}_m$) provided in Tables 3–7.

**Table 2.** Economic parameters for baseline POM design.

| Economic Parameters | Low Season | Medium Season | High Season | Average |
|---|---|---|---|---|
| Total capital costs, *CAPEX* (million US$) | | 18.42 | | 18.42 |
| Annualised *CAPEX* (million US$/y) | | 1.77 | | 1.77 |
| Labour costs, *LC* (million US$/y) | | 0.14 | | 0.14 |
| Total operating costs, *OPEX* (million US$/y) | 1.13 | 1.33 | 1.87 | 1.38 |
| Gross Profit, *GP* (million US$/y) | 3.89 | 5.64 | 8.10 | 5.53 |
| Economic Performance, *EP* (million US$/y) | 2.12 | 3.87 | 6.32 | 3.75 |

**Table 3.** Material and energy flows for baseline POM design.

| Material Flows | Low Season (t/h) | Low Season (t/y) | Medium Season (t/h) | Medium Season (t/y) | High Season (t/h) | High Season (t/y) | Average (t/h) | Average (t/y) |
|---|---|---|---|---|---|---|---|---|
| Low pressure steam, LPS | −15.3 | −66,600 | −20.3 | −88,300 | −28.8 | −125,300 | −20.3 | −88,500 |
| Utility water | −13.4 | −58,300 | −17.8 | −77,400 | −25.3 | −110,000 | −17.8 | −77,500 |
| Crude palm oil, CPO | 9.3 | 40,500 | 12.4 | 54,000 | 17.6 | 76,600 | 12.4 | 54,000 |
| Palm kernel, PK | 3.3 | 14,500 | 4.5 | 19,500 | 6.4 | 28,000 | 4.5 | 19,500 |
| Palm pressed fibre, PPF | 0 | 0 | 0 | 0 | 0 | 0 | 0 | 0 |
| Palm kernel shell, PKS | 1.4 | 6000 | 2.9 | 12,800 | 4.3 | 18,800 | 2.6 | 11,500 |
| Pressed empty fruit bunch, PEFB | 8.5 | 37,000 | 11.3 | 49,000 | 15.9 | 69,000 | 11.3 | 49,000 |
| Palm oil mill effluent, POME | 31.3 | 136,000 | 41.7 | 181,500 | 59.1 | 257,000 | 41.7 | 181,500 |
| **Energy Flows** | **(kW)** | **(MWh/y)** | **(kW)** | **(MWh/y)** | **(kW)** | **(MWh/y)** | **(kW)** | **(MWh/y)** |
| Electricity demand, $E^{\text{Demand}}$ (kW) | 990 | 4292 | 1100 | 4967 | 1600 | 6933 | 1200 | 5178 |

**Table 4.** Technology units operated for baseline POM design.

| Technology Units Operated (units) | Low Season | Medium Season | High Season |
|---|---|---|---|
| Tilted steriliser | 3 | 3 | 5 |
| Rotating drum separator | 1 | 2 | 2 |
| Oil pressing screw | 1 | 2 | 2 |
| Steam injection digester | 2 | 3 | 3 |
| Double screw press | 2 | 2 | 3 |
| Depricarper | 2 | 2 | 3 |
| Rolek nut cracker | 1 | 2 | 2 |
| Four-stage winnowing column | 1 | 1 | 1 |
| Vertical clarifier | 2 | 3 | 4 |
| Vacuum dryer | 2 | 2 | 3 |
| Three-phase decanter | 2 | 2 | 3 |
| Oil recovery pit | 1 | 2 | 2 |
| Water tube boiler | 1 | 2 | 2 |
| High-pressure steam turbine | 1 | 1 | 1 |
| Medium-pressure steam turbine | 2 | 2 | 3 |
| **Total** | **24** | **31** | **39** |

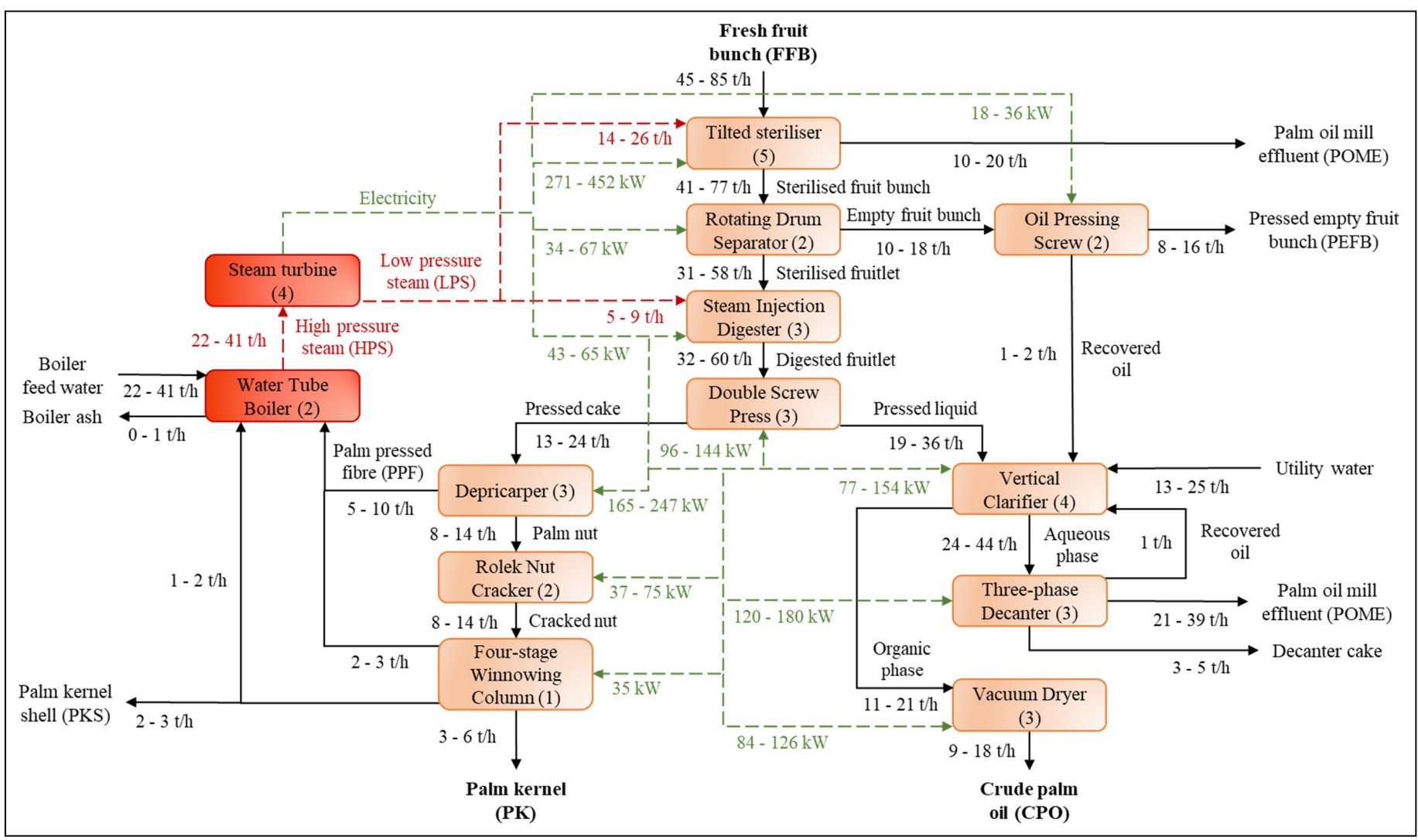

**Figure 3.** Baseline POM design.

**Table 5.** Process matrix **A** for palm oil milling process.

| | Material/Technology | te = 1 Tilted Steriliser (t/h) | te = 2 Rotating Drum Separator (t/h) | te = 3 Oil Pressing Screw (t/h) | te = 4 Steam Injection Digester (t/h) | te = 5 Double Screw Press (t/h) | te = 6 Depricarper (t/h) | te = 7 Rolek Nut Cracker (t/h) | te = 8 Four-Stage Winnowing Column (t/h) | te = 9 Vertical Clarifier (t/h) | te = 10 Oil Recovery (t/h) | te = 11 Vacuum Dryer (t/h) | te = 12 Three-Phase Decanter (t/h) | te = 13 Oil Recovery Pit (t/h) | te = 14 PPF Combustion (t/h) | te = 15 PKS Combustion (t/h) | te = 16 Water Tube Boiler (t/h) | te = 17 HPS Turbine (kW) | te = 18 MPS Turbine (kW) |
|---|---|---|---|---|---|---|---|---|---|---|---|---|---|---|---|---|---|---|---|
| m = 1 | Fresh fruit bunch, FFB (t) | −1 | 0 | 0 | 0 | 0 | 0 | 0 | 0 | 0 | 0 | 0 | 0 | 0 | 0 | 0 | 0 | 0 | 0 |
| m = 2 | Utility water (t) | 0 | 0 | 0 | 0 | 0 | 0 | 0 | 0 | −0.696 | 0 | 0 | 0 | 0 | 0 | 0 | 0 | 0 | 0 |
| m = 3 | Steam lost (t) | 0.12 | 0 | 0 | 0.116 | 0 | 0 | 0 | 0 | 0 | 0 | 0.138 | 0 | 0 | 0 | 0 | 0 | 0 | 0 |
| m = 4 | Sterilised fruit bunch (t) | 0.9 | −1 | 0 | 0 | 0 | 0 | 0 | 0 | 0 | 0 | 0 | 0 | 0 | 0 | 0 | 0 | 0 | 0 |
| m = 5 | Empty fruit bunch, EFB (t) | 0 | 0.24 | −1 | 0 | 0 | 0 | 0 | 0 | 0 | 0 | 0 | 0 | 0 | 0 | 0 | 0 | 0 | 0 |
| m = 6 | Sterilised fruitlet (t) | 0 | 0.76 | 0 | −1 | 0 | 0 | 0 | 0 | 0 | 0 | 0 | 0 | 0 | 0 | 0 | 0 | 0 | 0 |
| m = 7 | Digested fruitlet (t) | 0 | 0 | 0 | 1.04 | −1 | 0 | 0 | 0 | 0 | 0 | 0 | 0 | 0 | 0 | 0 | 0 | 0 | 0 |
| m = 8 | Pressed liquid (t) | 0 | 0 | 0 | 0 | 0.6 | 0 | 0 | 0 | -1 | 0 | 0 | 0 | 0 | 0 | 0 | 0 | 0 | 0 |
| m = 9 | Pressed cake (t) | 0 | 0 | 0 | 0 | 0.4 | −1 | 0 | 0 | 0 | 0 | 0 | 0 | 0 | 0 | 0 | 0 | 0 | 0 |
| m = 10 | Palm fruit nut (t) | 0 | 0 | 0 | 0 | 0 | 0.59 | −1 | 0 | 0 | 0 | 0 | 0 | 0 | 0 | 0 | 0 | 0 | 0 |
| m = 11 | Cracked nut (t) | 0 | 0 | 0 | 0 | 0 | 0 | 0.99 | −1 | 0 | 0 | 0 | 0 | 0 | 0 | 0 | 0 | 0 | 0 |
| m = 12 | Nut lost (t) | 0 | 0 | 0 | 0 | 0 | 0 | 0.01 | 0 | 0 | 0 | 0 | 0 | 0 | 0 | 0 | 0 | 0 | 0 |
| m = 13 | Aqueous phase (t) | 0 | 0 | 0 | 0 | 0 | 0 | 0 | 0 | 1.156 | 0 | 0 | −1 | 0 | 0 | 0 | 0 | 0 | 0 |
| m = 14 | Organic phase (t) | 0 | 0 | 0 | 0 | 0 | 0 | 0 | 0 | 0.54 | 0 | −1 | 0 | 0 | 0 | 0 | 0 | 0 | 0 |
| m = 15 | Palm pressed fibre, PPF (t) | 0 | 0 | 0 | 0 | 0 | 0.41 | 0 | 0.19 | 0 | 0 | 0 | 0 | 0 | −1 | 0 | 0 | 0 | 0 |
| m = 16 | Palm kernel shell, PKS (t) | 0 | 0 | 0 | 0 | 0 | 0 | 0 | 0.357 | 0 | 0 | 0 | 0 | 0 | 0 | −1 | 0 | 0 | 0 |
| m = 17 | Pressed empty fruit bunch, PEFB (t) | 0 | 0 | 0.868 | 0 | 0 | 0 | 0 | 0 | 0 | 0 | 0 | 0 | 0 | 0 | 0 | 0 | 0 | 0 |
| m = 18 | Decanter cake (t) | 0 | 0 | 0 | 0 | 0 | 0 | 0 | 0 | 0 | 0 | 0 | 0.113 | 0 | 0 | 0 | 0 | 0 | 0 |
| m = 19 | Crude palm oil, CPO (t) | 0 | 0 | 0 | 0 | 0 | 0 | 0 | 0 | 0 | 0.36 | 0.828 | 0 | 0 | 0 | 0 | 0 | 0 | 0 |
| m = 20 | Palm kernel, PK(t) | 0 | 0 | 0 | 0 | 0 | 0 | 0 | 0.453 | 0 | 0 | 0 | 0 | 0 | 0 | 0 | 0 | 0 | 0 |
| m = 21 | Recovered oil (t) | 0 | 0 | 0.132 | 0 | 0 | 0 | 0 | 0 | 0 | −1 | 0 | 0.02 | 0.0097 | 0 | 0 | 0 | 0 | 0 |
| m = 22 | Palm oil mill effluent, POME (t) | 0.23 | 0 | 0 | 0 | 0 | 0 | 0 | 0 | 0 | 0.64 | 0.034 | 0.867 | −1 | 0 | 0 | 0 | 0 | 0 |
| m = 23 | Deoiled POME (t) | 0 | 0 | 0 | 0 | 0 | 0 | 0 | 0 | 0 | 0 | 0 | 0 | 0.9903 | 0 | 0 | 0 | 0 | 0 |
| m = 24 | Boiler feed water (t) | 0 | 0 | 0 | 0 | 0 | 0 | 0 | 0 | 0 | 0 | 0 | 0 | 0 | 0 | 0 | −1 | 0 | 0 |
| m = 25 | Boiler ash (t) | 0 | 0 | 0 | 0 | 0 | 0 | 0 | 0 | 0 | 0 | 0 | 0 | 0 | 0.0423 | 0.039 | 0 | 0 | 0 |
| m = 26 | Low heating value (MJ) | 0 | 0 | 0 | 0 | 0 | 0 | 0 | 0 | 0 | 0 | 0 | 0 | 0 | 13388 | 17804 | −5151.8 | 0 | 0 |
| m = 27 | Low pressure steam, LPS (t) | −0.3 | 0 | 0 | −0.156 | 0 | 0 | 0 | 0 | 0 | 0 | 0 | 0 | 0 | 0 | 0 | 0 | 0 | 0.0316 |
| m = 28 | Medium pressure steam, MPS (t) | 0 | 0 | 0 | 0 | 0 | 0 | 0 | 0 | 0 | 0 | 0 | 0 | 0 | 0 | 0 | 0 | 0.0735 | −0.0316 |
| m = 29 | High pressure steam, HPS (t) | 0 | 0 | 0 | 0 | 0 | 0 | 0 | 0 | 0 | 0 | 0 | 0 | 0 | 0 | 0 | 1 | −0.0735 | 0 |
| m = 30 | Electricity (kW) | 0 | 0 | 0 | 0 | 0 | 0 | 0 | 0 | 0 | 0 | 0 | 0 | 0 | 0 | 0 | 0 | 1 | 1 |

**Table 6.** Technology specifications for palm oil milling process.

| Technology Specifications | $te = 1$ Tilted Steriliser | $te = 2$ Rotating Drum Separator | $te = 3$ Oil Pressing Screw | $te = 4$ Steam Injection Digester | $te = 5$ Double Screw Press | $te = 6$ Depricarper | $te = 7$ Rolek Nut Cracker | $te = 8$ Four-Stage Winnowing Column | $te = 9$ Vertical Clarifier | $te = 10$ Oil Recovery | $te = 11$ Vacuum Dryer | $te = 12$ Three-Phase Decanter | $te = 13$ Oil Recovery Pit | $te = 14$ PPF Combustion | $te = 15$ PKS Combustion | $te = 16$ Water Tube Boiler | $te = 17$ HPS Turbine | $te = 18$ MPS Turbine |
|---|---|---|---|---|---|---|---|---|---|---|---|---|---|---|---|---|---|---|
| Capacity, $\mathbf{CAP}_{te}$ (t/h.unit or kW/unit) | 20 | 50 | 10 | 20 | 25 | 10 | 10 | 15 | 10 | 100 | 8 | 20 | 41 | 100 | 100 | 25 | 1000 | 500 |
| Electricity, $\mathbf{E}_{te}$ (kW/unit) | 75.4 | 28 | 15 | 18 | 40 | 69 | 31 | 29 | 32 | 0 | 35 | 50 | 5.5 | 0 | 0 | 0 | 0 | 0 |
| Capital costs, $\mathbf{CC}_{te}$ (million US$/unit) | 1.2 | 0.23 | 0.12 | 0.15 | 0.18 | 0.25 | 0.18 | 0.25 | 0.15 | 0 | 0.39 | 0.30 | 0.03 | 0 | 0 | 2.00 | 0.83 | 0.61 |
| Operating costs, $\mathbf{OC}_{te}$ (million US$/unit.y) | 0.18 | 0.03 | 0.02 | 0.02 | 0.04 | 0.03 | 0.04 | 0.01 | 0.02 | 0 | 0.06 | 0.04 | 0 | 0 | 0 | 0.08 | 0.02 | 0.01 |

**Table 7.** Material costs, $\mathbf{C}_m$ for palm oil milling process.

| | $m = 1$ Fresh Fruit Bunch, FFB | $m = 2$ Utility Water | $m = 15$ Palm Pressed Fibre, PPF | $m = 16$ Palm Kernel Shell, PKS | $m = 16$ Palm Kernel Shell, PKS |
|---|---|---|---|---|---|
| **Material Costs, $C_m$ (US$/t)** | 121 | 0.55 | 23 | 45 | 45 |
| | $m = 17$ Pressed Empty Fruit Bunch, PEFB | $m = 18$ Decanter Cake | $m = 19$ Crude Palm Oil, CPO | $m = 20$ Palm Kernel, PK | $m = 24$ Boiler Feed Water |
| | 8 | 43 | 548 | 389 | 1.14 |

Intermediates associated with zero costs ($\mathbf{C}_m = 0$) are not listed here.

The assumption that the milling process can only be operated for 4350 h a year due to the working shifts of operators causes its capacity to be underutilised. In that case, more equipment units are required, resulting in greater *CAPEX* needed to process all the FFBs supplied, especially during the peak crop season. This shows a limitation in the previous study [6] during optimisation of a palm oil milling process. A more common practice in the industry is to increase the annual operating time, *AOT* of the process. In the industry, POM may operate up to 19 h/day or 7000 h/y ($AOT \leq 7000$). In that sense, the total capital costs, *CAPEX* needed can be reduced as lesser equipment units are required. However, the increment in *AOT* on top of 4350 h/y AST requires overtime cost; *OTC* paid for operators working extra time and operating costs, $\mathbf{OC}_{te}$ for service and maintenance of technology units. In this study, overtime costs, $C_{OT}$ of US$5/h and an additional 20% for $\mathbf{OC}_{te}$ are considered for operations exceeding 4350 h/y.

## 5. Results and Discussion

In order to achieve higher *EP*, an IOM was developed based on Equations (2)–(15) to optimise the baseline POM design with an objective function given in Equation (1). The model consists of 419 continuous variables with 54 integer variables and 622 constraints, solved in 17 s to achieve a global solution. The optimised POM design is presented in Figure 4 and the results (Table 8) showed that an *EP* of US$4.57 million/y *EP* is achieved (22% increment) as compared to US$3.75 million/y reported in the baseline design. This is mainly due to the reduction in *CAPEX* required, from US$18.42 to 11.56 million as the units of technology required, $\mathbf{U}_{te}^{\max}$ reduce from 39 (Table 4) to 26 units as shown in Table 9.

Data from Table 10 is compared with Table 3, showing that the same annual output is achieved, despite a smaller throughput (material flow per hour) in the optimised design by operating 5580, 5640, 4698 and 6656 h/y on average, low, medium and high crop seasons, respectively. In this respect, additional *OTC* by US$0.10, 0.03 and 0.17 million/y required for different crop seasons (an average of US$0.09 million/y). Besides, an additional 20% $\mathbf{OC}_{te}$ is required to operate the equipment due to longer operational time. However, *OPEX* is still reduced by US$0.32 million/y on average (= US$1.38–1.06 million/y) as the overall equipment operated decreases. It is worth mentioning that the equipment operated is the same for medium and high crop seasons but longer *AOT* in the latter case. As higher *OTC* is required to process all the fruits available during medium crop season with smaller processing capacity, it is more optimal to operate the milling process with higher throughput but lower *AOT*. Figure 5 shows the breakdown of costs allocation for both designs where *CAPEX* is annualised into a yearly basis for 15 years. It can be seen that the total costs required by the optimised design are lower than the baseline design by to 23% on average with 25, 18 and 25% reduction during low, medium and high crop seasons, respectively. In the next part of this section, the milling process is further analysed with FORA as mentioned earlier to study the feasible range of CPO output with respect to *CAPEX* invested.

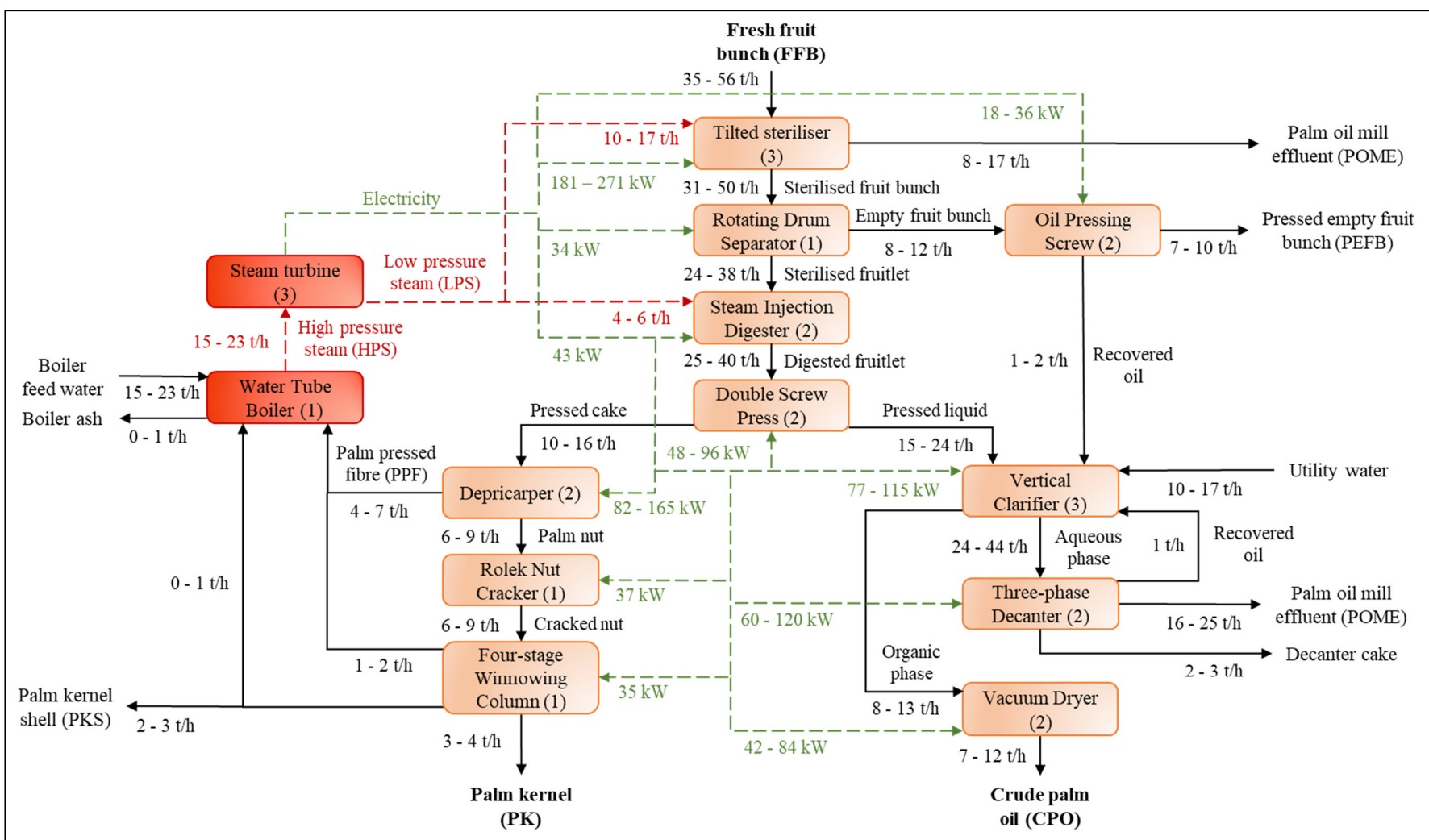

**Figure 4.** Optimised POM design.

**Table 8.** Economic parameters for optimised POM design.

| Economic Parameters | Low Season | Medium Season | High Season | Average |
|---|---|---|---|---|
| Annual operational time, *AOT* (h/y) | 5640 | 4700 | 6660 | 6520 |
| Total capital costs, *CAPEX* (million US$) | | 11.56 | | 11.56 |
| Annualised *CAPEX* (million US$/y) | | 1.11 | | 1.11 |
| Labour costs, *LC* (million US$/y) | | 0.14 | | 0.14 |
| Overtime costs, *OTC* (million US$/y) | 0.10 | 0.03 | 0.17 | 0.09 |
| Total operating costs, *OPEX* (million US$/y) | 0.94 | 1.40 | 1.40 | 1.21 |
| Gross Profit, *GP* (million US$/y) | 4.15 | 5.57 | 8.39 | 5.68 |
| Economic Performance, *EP* (million US$/y) | 3.04 | 4.46 | 7.28 | 4.57 |

**Table 9.** Technology units operated for optimised POM design.

| Technology Units Operated (Units) | Low Season | Medium Season | High Season |
|---|---|---|---|
| Tilted steriliser | 2 | 3 | 3 |
| Rotating drum separator | 1 | 1 | 1 |
| Oil pressing screw | 1 | 2 | 2 |
| Steam injection digester | 2 | 2 | 2 |
| Double screw press | 1 | 2 | 2 |
| Depricarper | 1 | 2 | 2 |
| Rolek nut cracker | 1 | 1 | 1 |
| Four-stage winnowing column | 1 | 1 | 1 |
| Vertical clarifier | 2 | 3 | 3 |
| Vacuum dryer | 1 | 2 | 2 |
| Three-phase decanter | 1 | 2 | 2 |
| Oil recovery pit | 1 | 1 | 1 |
| Water tube boiler | 1 | 1 | 1 |
| High-pressure steam turbine | 1 | 1 | 1 |
| Medium-pressure steam turbine | 1 | 2 | 2 |
| **Total** | 18 | 26 | 26 |

**Table 10.** Material and energy flows for optimised POM design.

| Material Flows | Low Season | | Medium Season | | High Season | | Average | |
|---|---|---|---|---|---|---|---|---|
| | (t/h) | (t/y) | (t/h) | (t/y) | (t/h) | (t/y) | (t/h) | (t/y) |
| Low pressure steam, LPS | −11.8 | −66,600 | −18.8 | −88,300 | −18.8 | −125,300 | −15.9 | −88,500 |
| Utility water | −10.3 | −58,300 | −16.5 | −77,400 | −16.5 | −110,000 | −13.9 | −77,500 |
| Crude palm oil, CPO | 7.2 | 40,500 | 11.5 | 54,000 | 11.5 | 76,600 | 9.7 | 54,000 |
| Palm kernel, PK | 2.6 | 14,500 | 4.2 | 19,500 | 4.2 | 28,000 | 3.5 | 19,500 |
| Palm pressed fibre, PPF | 0 | 0 | 0 | 0 | 0 | 0 | 0 | 0 |
| Palm kernel shell, PKS | 1.7 | 9500 | 1.7 | 7900 | 2.8 | 18,800 | 2.3 | 13,000 |
| Pressed empty fruit bunch, PEFB | 8.5 | 37,000 | 10.4 | 49,000 | 10.4 | 69,000 | 8.8 | 49,000 |
| Palm oil mill effluent, POME | 24.1 | 136,000 | 38.6 | 181,500 | 38.6 | 257,000 | 32.5 | 181,500 |
| **Energy Flows** | (kW) | (MWh/y) | (kW) | (MWh/y) | (kW) | (MWh/y) | (kW) | (MWh/y) |
| Electricity demand, $E^{\text{Demand}}$ (kW) | 660 | 3744 | 1000 | 4900 | 1000 | 6940 | 890 | 4938 |

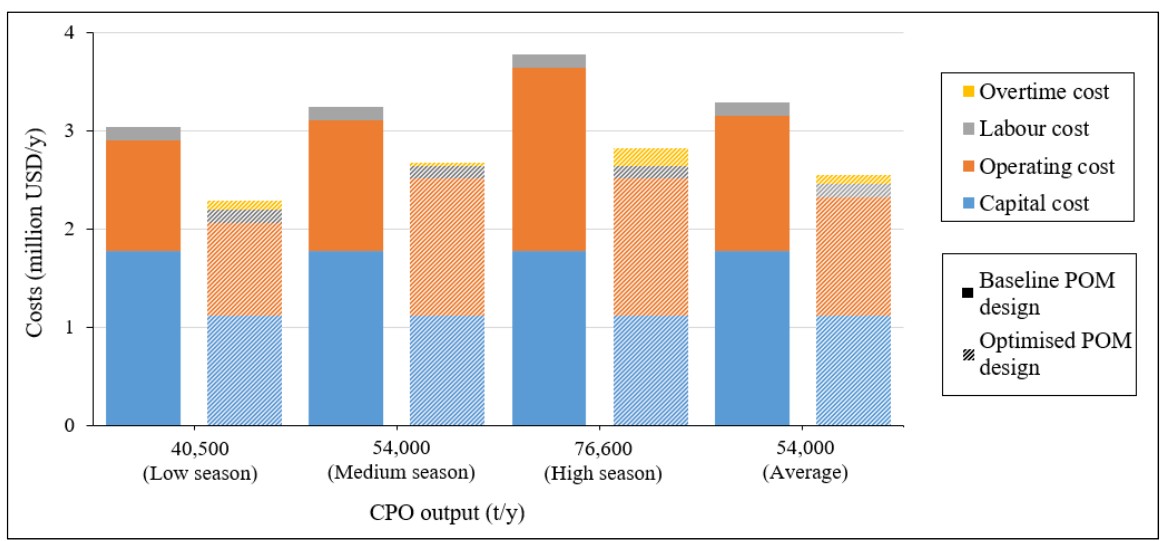

**Figure 5.** Costs allocation for a baseline against optimised palm oil mill design.

FORA is performed on the milling process based on Equations (16)–(18), subject to objective functions Equations (1) and (19) for each POM design while operational performance such as *UI* and *FI* are computed based on Equations (20)–(21). The analysis is performed for each increment in equipment unit added, beginning from the design with the smallest capacity of 18 units (optimised design during low crop season) to the design with the biggest capacity, 39 units in the baseline design. Graphical representations for FORA performed on different POM designs are presented in Figure 6 for different crop seasons and detailed information can be found in Table 11. From Figure 6a, we can see that the CPO production during low crop season, $(y_{CPO})_L$ falls within the entire feasible region, representing that each of the design can be used to achieve the output required. In this respect, the optimal operation will be determined from the trade-off between *OTC*, *CAPEX* and *OPEX* as a design with smaller capacity requires higher *OTC* but lower *CAPEX* and *OPEX* or vice versa. According to Tables 8–10, the POM is operated at smallest design capacity (*CAPEX* = US$8.36 million) with longer *AOT* of 5640 h/y (*OTC* = US$0.1 million/y) during the low crop season. However, the POM is operated in a different manner during medium crop season. Figure 6b shows that $(y_{CPO})_M$ lies in the feasible range for POM designs with 21 equipment units (*CAPEX* = US$9.18 million) and higher. Rather than operating the process with the smallest capacity possible, it was operated at a higher capacity (26 equipment units, *CAPEX* = US$11.56 million) due to lower *OTC* of US$0.03 million/y (*AOT* = 4700). On the other hand, at least 26 equipment units are needed during high crop season as $(y_{CPO})_H$ falls out of the feasible operating range for smaller POM design as shown in Figure 6c. From the optimised results, the smallest possible design with higher *OTC* of US$0.17 million/y (*AOT* = 6660) is operated during this crop season.

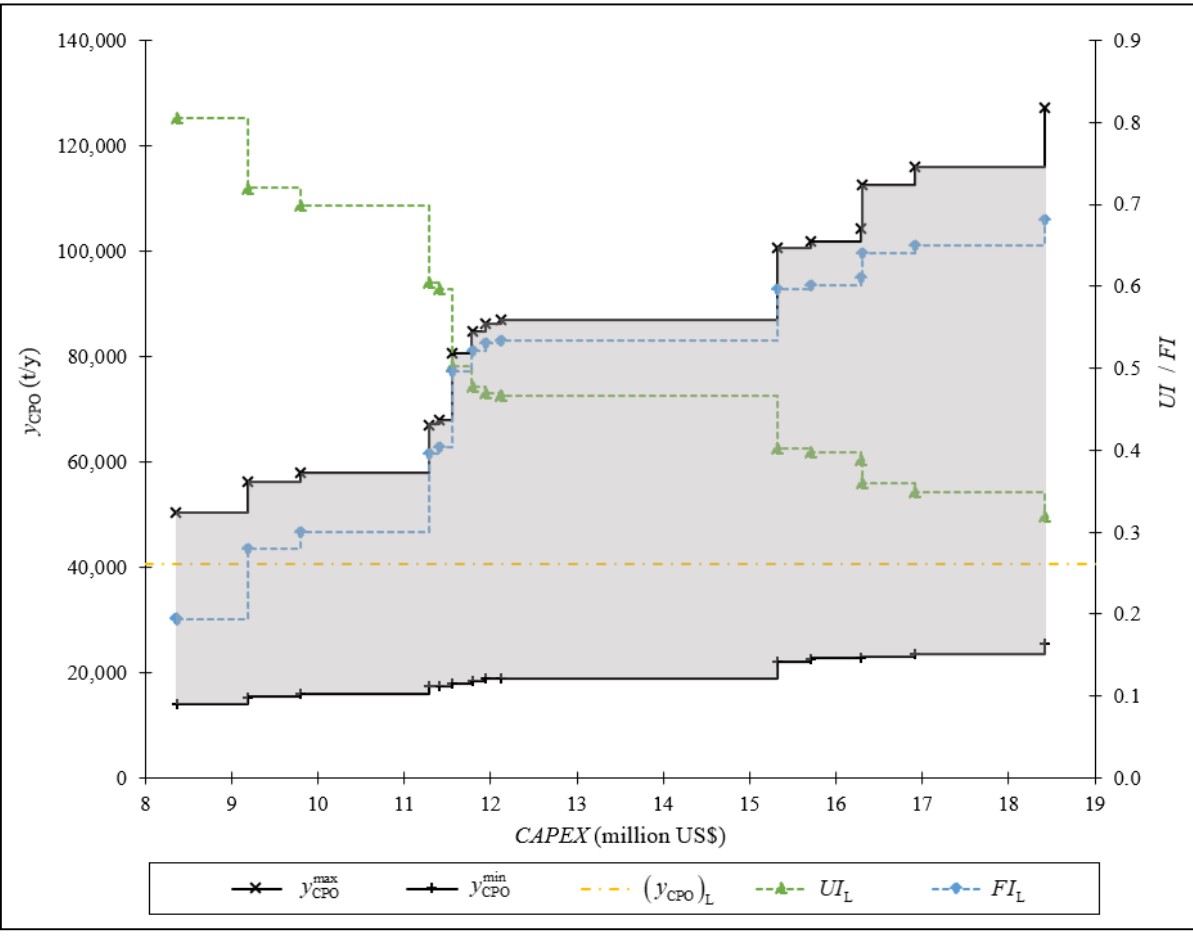

(**a**)

**Figure 6.** *Cont.*

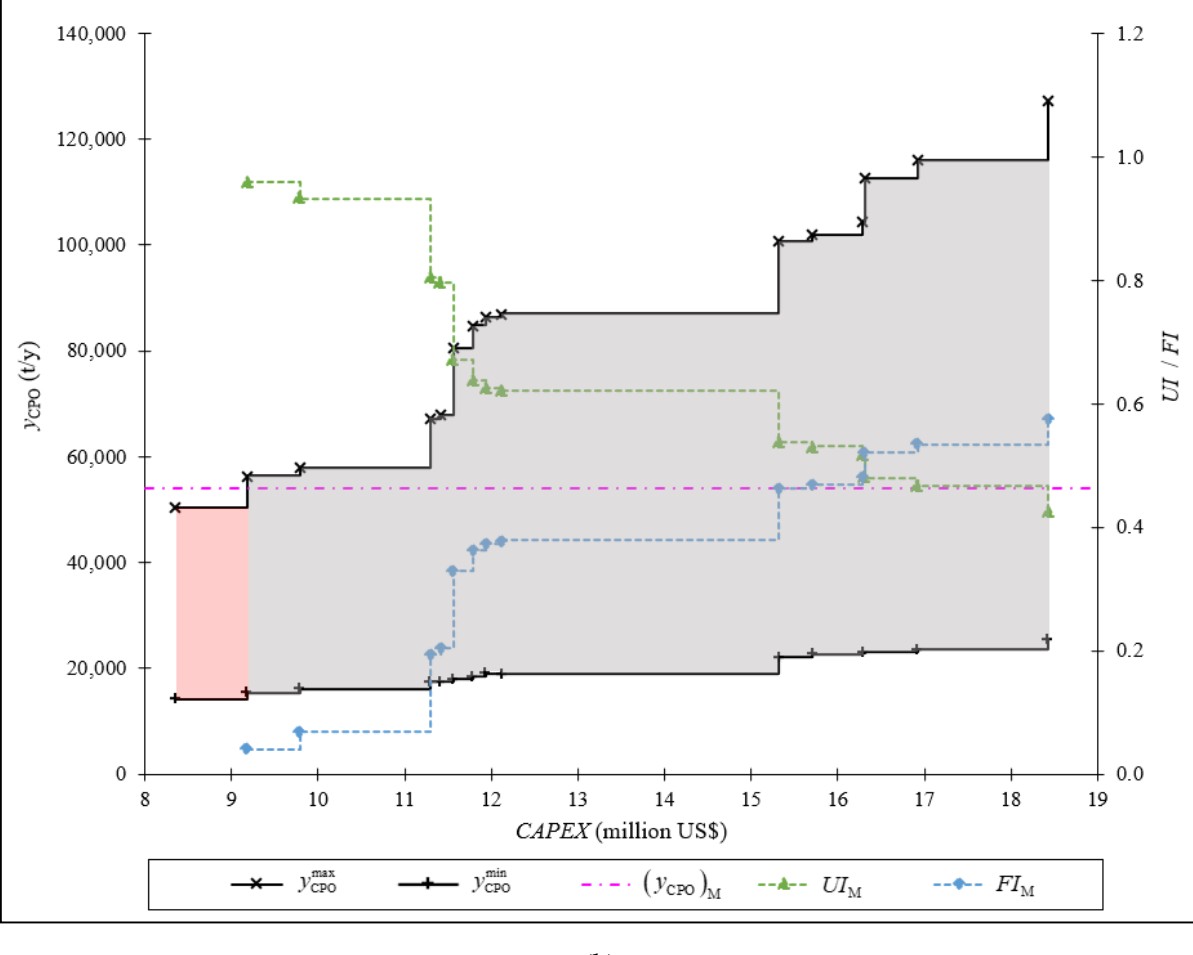

(**b**)

**Figure 6.** *Cont.*

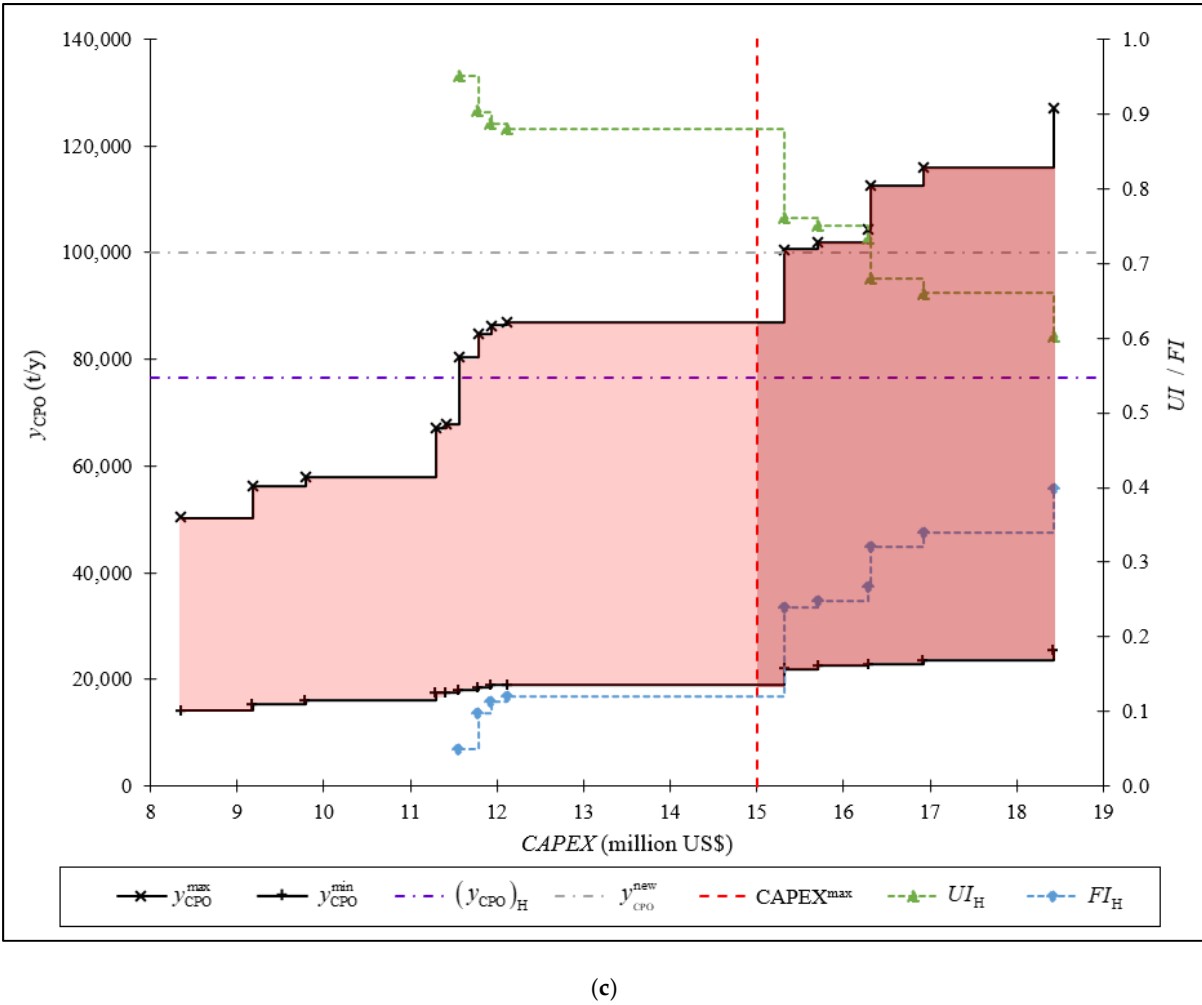

(**c**)

**Figure 6.** (**a**) Graphical representations for FORA performed during low crop season. (**b**) Graphical representations for FORA performed during medium crop season. (**c**) Graphical representations for FORA performed during high crop season with $y_{CPO}^{new}$.

**Table 11.** Feasible operating range analysis data.

| $U_{te}$ (units) | $y_{CPO}^{max}$ (t/y) | $y_{CPO}^{min}$ (t/y) | CAPEX (million US$) | Additional CAPEX (million US$) | Equipment Added | CBR | $UI_L$ | $FI_L$ | $UI_M$ | $FI_M$ | $UI_H$ | $FI_H$ |
|---|---|---|---|---|---|---|---|---|---|---|---|---|
| 18 | 50,300 | 14,100 | 8.36 | - | - | - | 0.81 | 0.19 | - | - | - | - |
| 21 | 56,300 | 15,400 | 9.18 | 0.82 | Vacuum dryer Double screw press | 3.5 | 0.72 | 0.28 | 0.96 | 0.04 | - | - |
| 22 | 58,000 | 16,100 | 9.79 | 0.61 | Depricarper MPS turbine | 2.9 | 0.70 | 0.30 | 0.93 | 0.07 | - | - |
| 24 | 67,000 | 17,500 | 11.29 | 1.50 | Three-phase decanter Tilted steriliser | 3.0 | 0.60 | 0.40 | 0.81 | 0.19 | - | - |
| 25 | 68,000 | 17,500 | 11.41 | 0.12 | Oil pressing screw | 3.1 | 0.60 | 0.40 | 0.80 | 0.20 | - | - |
| 26 | 80,500 | 18,000 | 11.56 | 0.15 | Steam injection digester | 44.3 | 0.50 | 0.50 | 0.67 | 0.33 | 0.95 | 0.05 |
| 27 | 84,800 | 18,500 | 11.79 | 0.23 | Rotating drum separator | 9.9 | 0.48 | 0.52 | 0.64 | 0.36 | 0.90 | 0.10 |
| 28 | 86,300 | 19,000 | 11.94 | 0.15 | Vertical clarifier | 5.5 | 0.47 | 0.53 | 0.63 | 0.37 | 0.89 | 0.11 |
| 29 | 87,000 | 19,000 | 12.12 | 0.18 | Rolek nut cracker | 1.3 | 0.47 | 0.53 | 0.62 | 0.38 | 0.88 | 0.12 |
| 31 | 100,600 | 22,000 | 15.32 | 3.21 | Tilted steriliser Water tube boiler | 2.5 | 0.40 | 0.60 | 0.54 | 0.46 | 0.76 | 0.24 |
| 32 | 101,900 | 22,700 | 15.49 | 0.18 | Double screw press | 2.7 | 0.40 | 0.60 | 0.53 | 0.47 | 0.75 | 0.25 |
| 35 | 104,400 | 22,900 | 16.29 | 0.80 | Depricarper Vacuum dryer | 1.6 | 0.39 | 0.61 | 0.52 | 0.48 | 0.73 | 0.27 |
| 36 | 112,600 | 23,000 | 16.31 | 0.02 | Vertical clarifier Oil recovery pit | 415.3 | 0.36 | 0.64 | 0.48 | 0.52 | 0.68 | 0.32 |
| 37 | 116,000 | 23,600 | 16.92 | 0.61 | MPS turbine | 5.9 | 0.35 | 0.65 | 0.47 | 0.53 | 0.66 | 0.34 |
| 39 | 127,200 | 25,500 | 18.42 | 1.50 | Three-phase decanter Tilted steriliser | 3.7 | 0.32 | 0.68 | 0.43 | 0.57 | 0.60 | 0.40 |

Apart from determining the feasible operating range of each design, this approach also serves as a tool to pinpoint the technology bottleneck, additional *CAPEX* needed and $y_{\text{CPO}}^{\text{max}}$ increment for the milling process in sequence. Table 11 shows that additional equipment units for vacuum dryer, double screw press and depricarper technologies are needed to increase $y_{\text{CPO}}^{\text{max}}$ from 50,300 to 56,300 t/y. Three different bottlenecks occur at the same time and $y_{\text{CPO}}^{\text{max}}$ can only be increased when all three equipment units added. It is then followed by MPS turbine and three-phase decanter to increase $y_{\text{CPO}}^{\text{max}}$ from 56,300 to 58,000 t/y and so on. It also allows the cost-benefit ratio, *CBR* for each step to be performed via Equation (22), providing more insight into the effectiveness of any additional investment made.

$$CBR = \frac{C_{\text{CPO}}\left(y_{\text{CPO}}^{\text{max}\,2} - y_{\text{CPO}}^{\text{max}\,1}\right)}{CRF(CAPEX_2 - CAPEX_1) + (OPEX_2 - OPEX_1)} \tag{22}$$

Based on the $y_{\text{CPO}}$ for each crop season, the *UI* and *FI* vary with its design. Note that *UI* and *FI* can only be measured when $y_{\text{CPO}}$ falls within the feasible operating range. During high crop season, a more significant portion of the production capacity in the optimised design has been utilised (*UI* = 0.95) as compared to the baseline design with *UI* of 0.60. However, it reduces the flexibility from *FI* of 0.40 to 0.05. This indicates that even though a higher proportion of the production capacity utilised in the optimised design during high crop season, the flexibility is reduced. In the event where $y_{\text{CPO}}$ were to increase further, it is implausible for the optimised design to cope up with such changes, unless, additional equipment units for rotating drum separator, vertical clarifier, rolek nut cracker and so forth are added. For instance, when $(y_{\text{CPO}})_{\text{H}}$ is increased by 30% from 76,600 to 100,000 t/y, *CAPEX* of US$15.32 million and 31 equipment units will be needed to achieve the $y_{\text{CPO}}^{\text{new}}$ as shown in Figure 6c. However, such increment could not be satisfied if $CAPEX^{\text{max}}$ is limited at US$15 million. Thus, a maximum of 87,000 t/y CPO could be produced with such given constraint in *CAPEX*.

## 6. Conclusions

A hybrid methodology was developed in this work to optimise a typical palm oil milling process to achieve maximum economic performance, performing an analysis for its operations and providing a feasibility study on the developed system. This hybrid approach consists of generic formulations for IOM and FORA to represent a palm oil milling process. The proposed approach has been illustrated using a Malaysian palm oil mill case study with multiple crop seasons. In the case study, higher *EP* is achieved from the optimised POM design with a smaller capacity but longer operational time as compared to the baseline design used. The utilisation of the POM has been improved. However, the flexibility of the process is also reduced proportionally. FORA serve as a decision-making tool to determine the *CAPEX* required, based on the output required with other constraints considered. Future research work will be directed to consider partial load models for changes in power consumption and process efficiency of each equipment units, analysing the detailed performance of each equipment and possibility for process intensification. Besides, sensitivity analysis for selling electricity, uncertainties in product prices and raw material availability due to external reasons and FORA formulation with multiple products output as well as the integration of downstream processes such as biorefinery can be included for decision-making in the future.

**Author Contributions:** Conceptualization, S.Z.Y.F. and D.K.S.N.; methodology, S.Z.Y.F.; software, S.Z.Y.F. and D.K.S.N.; validation, V.A., R.R.T. and D.C.Y.F.; formal analysis, S.Z.Y.F., V.A. and D.K.S.N.; investigation, R.R.T. and D.C.Y.F.; resources, D.K.S.N.; data curation, S.Z.Y.F.; writing—original draft preparation, S.Z.Y.F.; writing—review and editing, V.A., R.R.T. and D.C.Y.F.; visualisation, S.Z.Y.F.; supervision, D.K.S.N.; funding acquisition, D.K.S.N.

**Funding:** This research was funded by the Ministry of Higher Education, Malaysia through LRGS Grant (LRGS/2013/UKM-UNMC/PT/05).

**Acknowledgments:** Credit to Havys Oil Mill Sdn Bhd for the industrial data provided to develop an industrial case study in this work.

**Conflicts of Interest:** The authors declare no conflict of interest. The funders had no role in the design, analyses and interpretation of any data of the study.

## Nomenclature

*Abbreviation*

| | |
|---|---|
| CPKO | Crude Palm Kernel Oil |
| CPO | Crude Palm Oil |
| FFB | Fresh Fruit Bunch |
| FORA | Feasible Operating Range Analysis |
| IO | Input-output |
| IOM | Input-output Optimisation Model |
| PEFB | Pressed Empty Fruit Bunch |
| PK | Palm Kernel |
| PKS | Palm Kernel Shell |
| POM | Palm Oil Mill |
| POME | Palm Oil Mill Effluent |
| PPF | Palm Pressed Fibre |

*Sets*

| | |
|---|---|
| H | Index for high crop season |
| L | Index for low crop season |
| M | Index for medium crop season |
| $m$ | Index for material |
| $s$ | Index for crop season |
| $te$ | Index for technology |

*Variables*

| | |
|---|---|
| $AOT$ | Annual operational time |
| $\boldsymbol{B}_{te}$ | Technology $te$ bottleneck |
| $CAPEX$ | Total capital costs |
| $CAPEX_1$ | Total capital costs for design 1 |
| $CAPEX_2$ | Total capital costs for design 2 |
| $CAPEX_3$ | Total capital costs for design 3 |
| $CBR$ | Cost-benefit ratio |
| $CRF$ | Capital recovery factor |
| $E^{\text{Demand}}$ | Total electricity demand |
| $EP$ | Economic performance |
| $FI$ | Flexibility index |
| $GP$ | Total gross profit |
| $LC$ | Total labour costs |
| $OPEX$ | Total operating costs |
| $OTC$ | Total overtime costs |
| $UI$ | Utilisation index |
| $U_{te}$ | Number of equipment unit operated for technology $te$ |
| $\boldsymbol{x}_{te}$ | Processing capacity of technology $te$ |

| $x_{te}$ | Processing capacity of technology $te$ |
|---|---|
| $y_{CPO}$ | Crude palm oil output |
| $y_{CPO}^{new}$ | New crude palm oil output |
| $y_{electricity}$ | Electricity output |
| $y_{FFB}$ | Fresh fruit bunch input |
| $y_m$ | Input or output of material $m$ |
| $y_{CPO}^{max}$ | Maximum crude palm oil output |
| $y_m^{max}$ | Maximum input or output of material $m$ |
| $y_{m1}^{max}$ | Maximum input or output of material $m$ for design 1 |
| $y_{m2}^{max}$ | Maximum input or output of material $m$ for design 2 |
| $y_{m3}^{max}$ | Maximum input or output of material $m$ for design 3 |
| $y_{CPO}^{min}$ | Minimum crude palm oil output |
| $y_m^{min}$ | Minimum input or output of material $m$ |
| $y_{m1}^{min}$ | Minimum input or output of material $m$ for design 1 |
| $y_{m2}^{min}$ | Minimum input or output of material $m$ for design 2 |
| $y_{m3}^{min}$ | Minimum input or output of material $m$ for design 3 |
| $y_m^{new}$ | New input or output of material $m$ |

*Parameters*

| $\alpha_s$ | Fraction of occurrence for crop season $s$ |
|---|---|
| **A** | Matrix of material input and output ratios to and from technology $te$ |
| $a_{m,te}$ | Fixed interaction ratios between material m and technology $te$ |
| $AOT^{max}$ | Maximum annual operating time |
| AST | Annual shift time |
| $CAPEX^{max}$ | Maximum total capital costs |
| $\textbf{CAP}_{te}$ | Nominal capacity of technology $te$ |
| $\textbf{CAP}_{te}^{max}$ | Maximum capacity of technology $te$ |
| $(\textbf{CAP}_{te}^{max})^{-1}$ | Inverse matrix for maximum capacity of technology $te$ |
| $\textbf{CAP}_{te}^{-1}$ | Inverse matrix for nominal capacity of technology $te$ |
| $\textbf{CC}_{te}$ | Capital cost for technology $te$ |
| $C_{lab}$ | Cost of material $m$ |
| $\textbf{C}_m$ | Total overtime costs |
| $C_{OT}$ | Specific overtime cost |
| $\textbf{E}_{te}$ | Diagonal matrix for electricity consumption specified per unit technology $te$ |
| $FI^{new}$ | New flexibility index |
| $n_{wk}$ | Number of workers per shift |
| $n_{ws}$ | Number of working shifts per day |
| $\textbf{OC}_{te}$ | Operating and maintenance costs for technology $te$ |
| r | Discount rate |
| $t_{te}^{max}$ | Operational lifespan for technology $te$ |
| $\textbf{U}_{te}^{max}$ | Maximum units of technology $te$ installed |

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
