# Peer review of "Hybrid Approach for Optimisation and Analysis of Palm Oil Mill"

_processes, doi:10.3390/pr7020100_

Round 1
Reviewer 1 Report
Please find the attached file for the comments.

Author Response
We are very grateful for the prompt response. The feedback from each respective reviewer is highly appreciated. We have revised the manuscript according to the reviewers’ suggestions with the corrections as instructed. The changes made in the manuscript are summarised in the file attached,

Reviewer 2 Report
1. This manuscript describes two approaches, namely, input-output model (IOM) for optimization and feasible operating range analysis (FORA), as well as their application to palm oil mill (POM) processing. The two approaches are taken from the current literature but the case study is new. Overall, the case study is interesting and important, and the presentation is logical and detailed. The manuscript can be enhanced considering the following comments.2. The two approaches (i.e., IOM based optimization and FORA via graphical means) seem to be independent and can be performed separately. FORA may be using IOM but not optimization results. If so, what is the justification for using the phrase ‘hybrid approach’? What is the inter-dependency between the two approaches?
3. Lines 23 and 24: are the benefits compared to the base case (i.e., the case from the earlier work)? Clarify.
4. Lines 25 to 27: Add the additional capital required for the benefits stated in the last sentence of the abstract.
5. Some figures and tables are taken from reference 6. Do these require permission from the copyright owner (journal)?
6. Line 117: crap should be crop.
7. Eq. 1: state the variables, their type (real or integer) and bounds as well as constraints and their type (equality or inequality, linear or nonlinear), for maximizing EP.
8. Line 131: briefly define ‘variables’ and ‘parameters’ for clarity.
9. Eq. 19: since EP is minimized, can it become negative or zero? In other words, what is the lower bound for EP? Give the reasons in brief.
10. Is Figure 2 essential? Why not use a similar figure for the case study for explaining FORA?
11. Captions for figures should have more information. For example, Figures 2 and 6 have 4 plots each; briefly state what is presented in each of these plots.
12. Lines 254 to 255: is the optimization problem linear or nonlinear? Does it involve integers and continuous variables? Are multiple minima expected? Does LINGO provide the global solution always? If so, state the type of problems for which global solution is guaranteed. Why not use MS Excel (Solver or Premium Solver) instead of LINGO?
13. Line 270: discount rate (i.e., minimum acceptable rate of return) of 5% per annum is low. It should rather be about 10%.
14. How is ‘Average’ in Tables 2, 4, 8, 10 etc. calculated? Is it arithmetic average or weighted mean with fraction of occurrence of each season? Is the average EP of 3.75 in Table 2 correct?
15. Tables 3 and 10: Material flows in the middle (and not top) of the table should perhaps be Energy flows. Is it necessary to give material/energy flows in two different units (per hour and per year)?
16. Line 281: what/which reference is the ‘previous study’? Clarify.
17. Figures 3 and 4: what are the two numbers for the material/energy flows in these figures? Are they consistent with the material/energy flows in the corresponding tables?
18. Lines 331 to 333: why is FORA performed beginning from the design with the smallest capacity instead of the baseline design or optimized design?
19. Lines 349 to 350: how can you deduce from the results in Tables 9 and 10 that the design with smaller capacity is more optimal during the low crop season?
20. Is Figure 7 essential? Why not use a similar plot in Figure 6?
21. Journal name is missing in reference 6 (lines 485 to 486).
22. Is reference 22 published in the year 1936 available to authors? Did the authors see it?
Author Response
We are very grateful for the prompt response. The feedback from each respective reviewer is highly appreciated. We have revised the manuscript according to the reviewers’ suggestions with the corrections as instructed. The changes made in the manuscript are summarised in the file attached.

Round 2
Reviewer 2 Report
This is the revised manuscript in response to comments from Reviewer #2. The authors considered them, revised the manuscript suitably and provided their responses to each of the comments from Reviewer #2.
Text/numbers in a few figures and tables are too small. Ensure text/numbers in figures and tables are large enough for easy readability.